# Edge-Cloud Collaborative Defense against Backdoor Attacks in Federated Learning

**DOI:** 10.3390/s23031052

**Published:** 2023-01-17

**Authors:** Jie Yang, Jun Zheng, Haochen Wang, Jiaxing Li, Haipeng Sun, Weifeng Han, Nan Jiang, Yu-An Tan

**Affiliations:** 1Beijing Institute of Technology, School of Cyberspace Science and Technology, Beijing 100083, China; 2State Key Laboratory of Shield Machine and Boring Technology State Key Laboratory of Industrial Control Technology, China Railway Tunnel Bureau Group Co., Ltd., Zhengzhou 450000, China

**Keywords:** IoT, edge-cloud collaboration, federated learning, clean label attack

## Abstract

Federated learning has a distributed collaborative training mode, widely used in IoT scenarios of edge computing intelligent services. However, federated learning is vulnerable to malicious attacks, mainly backdoor attacks. Once an edge node implements a backdoor attack, the embedded backdoor mode will rapidly expand to all relevant edge nodes, which poses a considerable challenge to security-sensitive edge computing intelligent services. In the traditional edge collaborative backdoor defense method, only the cloud server is trusted by default. However, edge computing intelligent services have limited bandwidth and unstable network connections, which make it impossible for edge devices to retrain their models or update the global model. Therefore, it is crucial to detect whether the data of edge nodes are polluted in time. This paper proposes a layered defense framework for edge-computing intelligent services. At the edge, we combine the gradient rising strategy and attention self-distillation mechanism to maximize the correlation between edge device data and edge object categories and train a clean model as much as possible. On the server side, we first implement a two-layer backdoor detection mechanism to eliminate backdoor updates and use the attention self-distillation mechanism to restore the model performance. Our results show that the two-stage defense mode is more suitable for the security protection of edge computing intelligent services. It can not only weaken the effectiveness of the backdoor at the edge end but also conduct this defense at the server end, making the model more secure. The precision of our model on the main task is almost the same as that of the clean model.

## 1. Introduction

With the explosive development of mobile Internet and deep learning (DL), intelligent edge computing services based on collaborative learning are widely used in various application scenarios [1,2,3,4,5]. For example, autonomous vehicles and face recognition cameras; these intelligent services put forward higher requirements on the privacy and security of models [6,7,8]. Federated learning has the characteristics of distributed architecture and data out of the local area, which not only meet the privacy protection requirements of the Internet of Things but also obtain the collaboration model of the application [9,10,11].

However, in IoT applications, federated learning is vulnerable to malicious attacks [12,13], such as backdoor attacks [13,14,15,16,17,18]. Backdoor attacks during model training can skew the model to produce specific erroneous outputs on targeted inputs. The attacker manipulates the local client to plant the backdoor. Not only are the many other benign actors providing perfect cover for malicious actors, but the triggers embedded in the backdoor attacks are also very stealthy. Once the malicious model is uploaded to the global server, it will spread to all participants, posing a significant threat to the application of IIoT edge intelligent services [19]. For example, when a self-driving model encounters a traffic sign implanted with a rear door, it can easily recognize a “stop” sign as a “go” sign, which will lead to catastrophic traffic accidents. As shown in Figure 1, when implementing a backdoor attack on an intelligent edge computing service, the adversary embeds the local model of one of the edge nodes into the backdoor trigger through training samples. After aggregating and synchronizing the models, the embedded attack patterns are propagated to all edge nodes.

Many scholars try to prevent backdoor attacks by limiting the impact of global model poisoning updates [20] or detecting triggers [15] to clean malicious model updates [21,22,23]. However, federated edge computing intelligent service applications may face more difficulties. Bandwidth constraints and unstable network connections prevent IIoT innovative applications from retraining local models or downloading the latest global models. Therefore, we believe it is more realistic to directly mitigate or eliminate the threat of malicious models to victim applications.

In this paper, we focus on mitigating backdoor attacks by weakening the impact of malicious models on innovative service clients. Based on the actual edge-cloud collaborative intelligent manufacturing scenario, we propose an edge-cloud collaborative backdoor defense framework, which ensures the accuracy of the main task of the model while minimizing the success rate of backdoor attacks. On the one hand, backdoors are detected and eliminated at edge nodes, and model parameters are trained as cleanly as possible to ensure the security of edge models. On the other hand, the cloud server judges whether the current input is an honest update vector by using the cosine similarity between the local update vector of the current round and the global update result of the previous round, that is, using the current input and the previous results.

The main contributions and innovations are as follows:We mainly remove the correlation between the backdoor samples and the attack target at the edge and repair the model accuracy by self-distillation to resist the backdoor attack of the edge intelligent service.Our framework is more suitable for the edge collaborative computing scenario of intelligent services.We propose the idea of a layered defense backdoor to obtain the cooperative training model more safely.

The organizational structure of the paper is as follows. Section 2 discusses the related work starting with related work on FL followed by backdoor attack and defense. Section 3 presents the threat model. Section 4 describes the detail the layered rear door defense framework we proposed. Section 5 describes the experimental environment and the evaluation of experimental results. The last section presents the conclusion and future work.

## 2. Related Work

This section first introduces the working principle of federated learning and gives the training process. Then, it describes the classification and methods of federated backdoor attacks. At last, we introduce the existing techniques of edge coordination for federal backdoor defense.

### 2.1. Federated Learning

FL data—as a distributed paradigm—do not leave the local area, protects the critical privacy of participants through interactive model updates [24]. In the edge computing intelligent service scenario, the FL framework consists of *n* edge nodes and a cloud server. Considering that all the training data of the FL system are *D*, each edge node has its own training data set $D_{i}$. The ith edge node is $D_{i}$, $D=D_{1},D_{2},\cdots ,D_{N}$, Di has a dataset size of $L_{i}$, $L_{i}=D_{i}$, and the total number of trained samples is $L=\left|D\right|=\sum_{i}^{N}L_{i}$. The training data in FL are stored and processed only on the edge node. The specific process of federal learning and training is shown in Figure 2.

When the FL system is initialized, the cloud aggregator initializes a global model $w^{0}$ and distributes it to randomly selected edge nodes for local training. In round *t*, the server randomly selects *m* edge nodes. Each selected edge node shares the global model parameter $w_{g}^{t}$, train the model on its local data to obtain a new local model weight $W_{L}^{t+1}$, and send it to the cloud server for aggregation to obtain a new global model parameter $w_{g}^{t+1}$.

Specifically, at the t−th iteration, the FL system repeats the following three steps, obtaining the new $w_{g}^{t+1}$ based on the current global model $w_{g}^{t}$.

Step 1: the cloud server sends the global model $w_{g}^{t}$ to randomly selected edge nodes.

Step 2: the selected edge node uses local data and global model $w_{g}^{t}$, train the local model, and generate a new local model parameter $w_{l}^{t+1}$ through the random gradient descent method.

Step 3. The cloud server uses the standard FedAvg aggregation rules to aggregate the selected local model updates to obtain the next round of model $W_{g}^{t+1}$.
(1)$$\begin{array}{c} W_{t+1}=W_{i}^{t}+\frac{\eta }{N}\sum_{i=1}^{N}W_{i}^{t+1}-W_{g}^{t}, \end{array}$$
where $\frac{\eta }{N}$ is the task-specific global learning rate set by the server.

### 2.2. FL Backdoor Attack

In federated learning, attackers usually perform backdoor attacks by invading the training data of the client and updating the local model. The injected backdoor will not affect the model’s behavior on the clean sample but will force the model to generate unexpected behavior when adding a specific trigger to the input [15,25,26]. Bagdasaryan et al. [14] proposed that only one attacker needs to be selected in the federal learning and training process, and only 1% of the backdoor data are used to make the global model achieve 100% accuracy on the backdoor task. A distributed backdoor attack is a new attack method proposed in [27], where multiple attackers cooperate to implant the backdoor.

Because there are many clients in federated learning, the contribution of the back door model will be diluted when the server aggregates. Therefore, it is necessary to enlarge and modify the parameters of the trained malicious model. These methods also eliminate the updates of benign clients with large deviations, resulting in poor performance of the aggregated global model for such clients.

For backdoor attacks on local models, attackers are generally given greater rights to update the model of the selected client directly. Backdoor attacks are generally performed on the training process or the trained model. Many studies show that the effect of model poisoning is better than that of data poisoning, and the impact of the distributed backdoor attack is better than that of a centralized backdoor attack. The distributed backdoor attack is to split the global trigger into local triggers. Multiple attackers use different local triggers to attack the FL, and finally, all the motivations of the local model in the server from the triggers of the global model. Such a backdoor is more concealed and more aggressive.

### 2.3. FL Backdoor Defense

In the edge-cloud collaborative federated learning framework, attackers often implement targeted poisoning attacks, including backdoor attacks by a single attacker and Sybil attacks [28] in which multiple attackers conspire. For different attack methods, scholars have given many targeted defense methods [29,30]. In backdoor attacks, defenders often defend at different stages. For example, during client training, Hou et al. [31] proposed a defense method based on backdoor area filtering. The interpretable AI model is used on the server to build multiple filters and send them to the client randomly to prevent advanced attackers from escaping defense. For the input determined as the back door, the client uses the fuzzy and label flipping strategies to clear the back door trigger area on the data and restore the availability of the data.

Before local model aggregation, Nguyen et al. [32] proposed an adaptive defense method based on HDBSCAN clustering, clipping, and noise adding. The author combines the detection of abnormal model updates with the tailoring of weights. Firstly, the local updates with high backdoor effects are removed, and then the residual backdoors are eliminated by tailoring and adding noise. Finally, the clipping factor and the amount of noise required to remove the backdoor are minimized to maintain the benign performance of the global model. Gao et al. [33] proposed an aggregation rule PartFedAvg, which limits the upload ratio of model updates. By designing parameter *d* to control the uploading ratio of each client model update, the malicious client cannot upload complete malicious parameters quickly, making it difficult to carry out backdoor attacks.

In local gradient aggregation, Mi et al. [34] propose a defense method based on Mahalanobis distance similarity. The authors perform backdoor attacks assuming that the scores of benign clients will be distributed within a small range, while malicious clients have significantly higher or lower scores. The authors first zero-center preprocess the gradients of the filters in a layer of the CNN, then employ unsupervised anomaly detection to evaluate the preprocessed filters and compute an anomaly score for each client. Finally, it is determined whether it is a malicious client according to its abnormal score. Most benign clients’ scores will be distributed in a small range, while malicious clients should have significantly higher or lower scores. In the FL aggregation protocol, the server cannot know the plaintext of the user’s local model parameters, and the server cannot detect the abnormal contribution of the participant to the global model.

After local gradient aggregation, Wu et al. [35] proposed a distillation-based federated forgetting method to resist backdoor attacks. By subtracting the target client’s accumulated historical updates from the global model to remove its contribution, the old global model is used as the teacher model to train the forgetting model. The model’s performance is recovered through knowledge distillation. Zhao et al. [36] Proposed how to realize the detection and defense of backdoor attacks of Federated learning from the perspective of combining participants and servers of Federated learning.

In the Sybil attack, the attacker disguises multiple participants, eventually leading to the federated learning model and a significant reduction in the model effect. To combat Sybil attacks, Fang et al. [28] proposed a FoolsGold defense mechanism to mitigate Sybil attacks. When FoolsGold aggregates model updates submitted by participants, it reduces the weight of participants with similar model updates while keeping the weight of participants with different model updates unchanged.

This paper mainly focuses on defense mechanisms against backdoor attacks applicable to Internet scenarios such as edge computing and intelligent services.

## 3. Problem Setting and Objectives

This section mainly details the problems to be solved in this paper, the enemy’s attack targets and attack capabilities, the defender’s defense targets and capabilities, and defense evaluation measures.

### 3.1. Problem Definition

To deal with the back door attack of edge cloud collaboration of intelligent manufacturing and edge computing intelligent services, we propose the idea of layered back door defense. Assuming that the total number of training periods is T, we divide the whole training process into two stages: edge defense and cloud service defense.

At the edge device, we assume that the backdoor opponent has generated a set of backdoor examples in advance and successfully injected them into the training dataset. The defender does not consider whether the data are poisoned or not and does not know the proportion and distribution of backdoor instances in a given dataset. Defenders expect to train clean models and models trained on clean data.

Consider a classification task with a dataset $D=D_{c}\cup D_{b}$, where $D_{c}$ denotes the subset of clean data and $D_{b}$ denotes the subset of backdoor data. Training trains a DNN model $f_{\theta }$ by minimizing the empirical error:(2)$$\begin{array}{c} £=E_{\left(x,y\right)∼D_{c}}\left[£\left(f_{\theta }\left(x\right),y\right)\right]+E_{\left(x,y\right)∼D_{b}}\left[£\left(f_{\theta }\left(x\right),y\right)\right], \end{array}$$
where £(·) represents the loss function of cross entropy loss. The task at the whole edge is decomposed into the clean data Dc task and the backdoor task of the backdoor data Db, thus generating a backdoor model.

To prevent backdoor attacks, we minimize the backdoor learning experience error:(3)$$\begin{array}{c} £=minE_{\left(x,y\right)∼D_{c}}\left[£\left(f_{\theta }\left(x\right),y\right)\right]+E_{\left(x,y\right)∼D_{b}}\left[£\left(f_{\theta }\left(x\right),y\right)\right]. \end{array}$$

On the server side, the defender does not detect or identify backdoor model updates. Whether a model update is poisoned or not, the defender must implement a self distilling countermeasure.

### 3.2. Threat Model

We use a typical FL setup where multiple edge nodes use the FedAvg algorithm in a cloud service to train the ML model collaboratively. We also assume that public datasets are available for tuning cloud server models. This assumption is generally applicable in practice. Typical datasets are essential when designing neural network structures in FL.

We consider the backdoor data of some attackers with $K^{\prime }\geq K/5$, *K* not exceeding 20% of the staff, and poisoning data not exceeding 10%. However, it is unlikely that there are more than 20% attackers in the real FL scene. In addition, we assume that edge nodes and FL servers are honest and uncompromising and that attackers cannot control aggregators or honest workers. It is different from the previous ideas and more in line with intelligent manufacturing requirements. We ensure that the data handed over at any stage are safe and reliable.

Adversary goals. First, make sure that the model’s main task is correct. Second, ensure a high success rate of backdoor mission attacks. Finally, the concealment is high. The following formula can summarize the attacker’s goal. That is, the attacker manipulates the model to give the specified class.
(4)$$\begin{array}{c} f\left(G^{{}^{\prime }},x\right)=\left\{\begin{array}{cc} & c^{{}^{\prime }}\neq f\left(G,x\right)\;\mathrm{if}\;\forall x\in I_{A}\\ & f\left(G,x\right)\;\mathrm{if}\;x\notin I_{A}. \end{array}\right. \end{array}$$

To be more stealthy, it is necessary to make it difficult to distinguish the poisoning model from the benign model as much as possible. An attacker can estimate this distance by comparing the local malicious model with the global or local model trained based on benign data.
(5)$$\begin{array}{c} {£}_{attacker}=\alpha {£}_{class}+\left(1-\alpha \right){£}_{ano}, \end{array}$$
where ${£}_{class}$ is the target classification of the main task mentioned in the context, ${£}_{ano}$ is the deviation of the malicious model from the benign model, and the parameter α is used to control the trade-off between attack concealment and aggressiveness.

Adversarial capabilities. Traditional edge-cloud-based federated learning backdoor defense methods mostly assume that edge devices are untrustworthy and only defend against backdoors after model aggregation. In the edge computing scenario, there are relatively few participants in collaborative learning, and the effect of backdoor attacks will be more prominent. Therefore, it is more practical to directly reduce the impact of malicious clients on the global model and even benign clients on edge devices.

Defense’s goals. In the edge computing intelligent service scenario, the edge node detects the data uploaded by the client in time and trains a clean model as much as possible. The cloud server must defend the model update uploaded by edge nodes and the aggregated global model again. It is a reasonable defense for realistic scenarios. Edge computing intelligent services collaborative computing requires ensuring that every step is safe and correct. We verify whether the global model is attacked in stages and reduce the success rate of backdoor attacks without affecting the accuracy of the main task. The defender’s goal is to have a model $f_{\theta^{*}}$ designed to be immune to backdoor attacks, i.e.,
(6)$$\begin{array}{c} f_{\theta^{*}}\left(x+\delta \right)=f_{\theta^{*}}\left(x\right). \end{array}$$

Defender’s capabilities. On the server side, defenders cannot violate the privacy-preserving principles of federated learning. The server cannot access each local agent’s training data or training process. The server cannot directly access the local model parameters or updates of the client and can only have a small group of clean validation data sets.

Evaluation Metrics. We use two metrics to evaluate: (1) Attack success rate: mainly by judging the classification confidence of the target image in the global model, that is, classifying the target sample into the specified category. It corresponds to Equation (Equation 3), where $L_{adv}\left(\theta \right)$ denotes the adversarial loss, $y_{adv}$ denotes the target label; (2) Basic task accuracy: the global model should have high accuracy for non-target samples. It corresponds to Equation (Equation 4), where L(θ) denote normal training loss, y denotes a real label.
(7)$$\begin{array}{c} L_{adv}\left(\theta \right)=:L\left(F\left(x^{t},\theta \right),y^{adv}\right) \end{array}$$
(8)$$\begin{array}{c} L\left(\theta \right)=:\frac{1}{N}\sum_{i=1}^{n}L(\left(x_{i},\theta \right),y). \end{array}$$

## 4. Proposed Method

Based on the problem definition and the defender’s goals, we describe a federated learning backdoor defense approach for IoT edge collaboration in this section. To better apply our method to the Internet of Things scenarios of intelligent edge computing services and to better meet the real-time protection requirements of managers, we set up real-time defense methods on edge devices and set forgetting learning. The distillation mechanism makes each round of federation cooperation training safe and reliable.

### 4.1. Overview

This section describes the proposed solution: a layered backdoor defense framework based on edge computing imaginative service scenarios. The concept of hierarchical governance derives from the IoT edge cloud collaboration scenario. As IIoT faces increasingly complex issues—for example, limited bandwidth, and unstable network connections—edge devices cannot retrain their models or update global models promptly. Therefore, according to the actual situation, it is more practical to adopt the idea of hierarchical governance to directly mitigate the impact of malicious clients on the global model or benign clients on edge devices. Layered defense is a hybrid solution of centralized and distributed defense approaches. On the one hand, layered defenses can handle the fault-tolerance limitations of centralized defenses in complex IoT. On the other hand, the layered defense can address distributed defense’s fully decentralized management challenges. In this article, no restrictions are imposed on the attacker. Attackers can perform backdoor attacks at any time and any stage. To ensure the accountability of all parties, we propose the idea of a layered defense.

Our defense mainly includes edge defense and cloud defense. On the edge side, detect and eliminate the impact of edge backdoors and train as clean model parameters as possible to ensure the accuracy of model updates uploaded to the cloud server. Before sending model updates to local edge nodes, we perform an edge self-distillation mechanism to improve model security and accuracy further. On the server side, judge whether the current input is an honest vector between the current round of local updates and the previous round of global updates, and then judge the dotted line similarity between the model updates to detect the backdoor model update, and finally, using the self-attention distillation mechanism to restore model performance.

### 4.2. Backdoor Defense Based on Edge Computing Services

We focus on classification tasks using deep neural networks. Backdoor defense based on edge computing services, introducing edge computing defense and cloud service defense, respectively.

Edge defense.According to the experiment, we know that the model learns the backdoor data much faster than the clean data, and the stronger the attack, the faster the model converges on the backdoor data, as shown in Figure 3.

Literature ABL [37] also confirmed this point of view. In order to defend against backdoor attacks in federated learning, we need to prevent this backdoor learning capability. In this paper, we use the method with the largest loss value to detect backdoor samples, use forgetting learning to forget backdoor samples, break the correlation between backdoor samples and target classes, and train a clean local model as much as possible. Finally, to make the main task’s accuracy unaffected, we propose an attentional self-distillation method to recover the model performance. First, using the idea of ABL for reference, backdoor samples are detected by maximizing the loss value. The first feature of backdoor attacks is combined with the Local Gradient Ascension Mechanism (LGA) to isolate a portion of backdoor samples with large loss values. Then a global gradient ascent strategy (GGA) is utilized to break the strong correlation between backdoor samples and backdoor target classes. The specific formula is as follows:(9)$$\begin{array}{c} {£}_{ABL}^{t}=\left\{\begin{array}{cc} & {£}_{ABL}=E_{(x,y)∼D}\left[sign\left(£\left(f_{\theta }\left(x\right),y\right)-\gamma \right)\cdot £\left(f_{\theta }\left(x\right),y\right)\right]\;\mathrm{if}\;0\leq t<T_{te}\\ & {£}_{GGA}=E_{\left(x,y\right)∼\hat{D_{c}}}\left[£\left(f_{\theta }\left(x\right),y\right)\right]-E_{\left(x,y\right)∼\hat{D_{b}}}\left[£\left(f_{\theta }\left(x\right),y\right)\right]\;\mathrm{if}\;T_{te}⩽t<T. \end{array}\right. \end{array}$$

LGA can retain the difference between clean and backdoor samples for some time and then successfully screen backdoor samples. GGA treats the backdoor task as a dual-task learning process, minimizing the clean sample inference loss and maximizing the backdoor sample inference loss.

Edge distillation. Even if a small number of backdoor samples bypass the defense mechanism of edge nodes, we can defend against them at this stage. Instead of directly using the trained edge service model ${£}_{model}$ parameters as the final model of this round, we use Self Attention Distillation (SAD) [38] to fine-tune it to obtain a shadow model ${£}_{model}^{{}^{\prime }}$. Attention self-distillation improves the model’s accuracy and can defend against more stealthy backdoors. The Self Attention Distillation model can learn from itself and improve substantially without additional supervision or labels. Since self-distillation provides a single neural network executable of varying depths, it allows adaptive accuracy and efficiency trade-offs on resource-constrained edge devices. Therefore, we propose an attention self-distillation mechanism for edge models at edge nodes.

Specific steps are as follows: Firstly, we fine-tune the local edge model ${£}_{model}$ to obtain a fine-tuned shadow model ${£}_{model}^{{}^{\prime }}$. Secondly, the attention map operator is used to calculate the output of each activation layer of the shadow model and map it to the attention map. Then, the shadow model is self-distilled layer by layer using the attention feature map, and the distillation loss between layers is obtained. Finally, a high-precision boosted model is obtained with the model’s cross-entropy loss function training.

Attention Representation. $A_{m}\in R^{C_{m}\times H_{m}\times W_{m}}$ denotes the activation output of the mth layer of the network, where $C_{m}$,$H_{m}$ and $W_{m}$ represent the channel, height, and width, respectively. Note that the generation of the graph is equivalent to finding a mapping function *g*: $R^{C_{m}\times H_{m}\times W_{m}}\to R^{H_{m}\times W_{m}}$ The absolute value of each element in this map represents the importance of that element to the final output. So the mapping function can be constructed from the computed statistics of these values across channel dimensions. Specifically, it can be used as a mapping function by operating Formula (1):(10)$$\begin{array}{c} G_{p}^{sum}\left(A_{m}\right)=\sum_{i=1}^{C_{m}}{\left|A_{mi}\right|}^{p} \end{array}$$
where p>1 and Ami represent the ith slice of Am in the channel dimension.

On $G_{2}^{sum}\left(A_{m}\right)$ Execute space softmax operation on Φ(:). If the original attention map size is different from the target, add bilinear upsampling B(:) before the softmax operation.

Attention Distillation Loss. Attention Distillation Loss. The continuous layered distillation loss formula is as follows:(11)$$\begin{array}{c} {£}_{distill}\left(A_{m},A_{m+1}\right)=\sum_{M-1}^{m=1}{£}_{d}\left(\Psi \left(A_{m}\right),\Psi \left(A_{m+1}\right)\right). \end{array}$$
$L_{d}$ is usually defined as the L2 loss. $\Psi (A_{m})$ is the target of distillation loss.

Overall Training Loss. The overall training loss is a combination of the cross entropy (CE) loss and continuous layered distillation losses.
(12)$$\begin{array}{c} {£}_{total}={£}_{seg}\left(s,\hat{s}\right)+\beta {£}_{exist}\left(b,\hat{b}\right)+\delta {£}_{distill}\left(A_{m},A_{m+1}\right). \end{array}$$

The first two losses are segmentation losses, including the standard cross-entropy loss ${£}_{seg}$. The purpose of the IoU loss is to discriminate the degree of correlation between the prediction results and the detection results. ${£}_{exist}\left(b,\hat{b}\right)$ is a binomial cross-entropy loss function. ${£}_{distill}\left(A_{m},A_{m+1}\right)$ is the distillation loss function. *s* represents the actual classification situation, $\hat{s}$ is the predicted classification result, *b* represents the existence of the category, $\hat{b}$ represents the existence of the model prediction result, $A_{m}$, $A_{m+1}$ is the activation area, and β, δ are the adjustment of the balance loss on the final task.

### 4.3. Cloud Service Defense

On the server side, comparing the local update vector of the current round with the global update result of the previous round is equivalent to judging the following data based on previous experience. Use the relu strategy to remove outliers and obtain the remaining *L* benign models. Then, calculate the cosine similarity between pairs of L vector models, and use the outlier detection method to regard more than 50% of the classes as benign updates and the others as outliers, and obtain *m* benign models. Finally, a self-distillation strategy is performed on *m* benign models to repair the accuracy of the model. After the model updates of edge devices are aggregated, we fine-tune the model using a small portion of the clean test set, using the self-distillation method of the attention mechanism to recover the model performance. We first fine-tune the aggregated global model to obtain a fine-tuned model. Then, use the attention map operator to calculate the output of each activation layer of the shadow model and map it to the attention map. The layer-by-layer self-extraction of the shadow model is performed using the attention feature map, and the distillation loss between layers is obtained. Finally, combined with the cross-entropy loss function training of the model, a safe and high-precision merged model is obtained and the model parameters are sent to the edge server in the next round. We use a clean test set to train a fine-tuned shadow model in this part.

## 5. Experimental Evaluation

In this section, we have verified the effectiveness of our method in various ways. First, the experimental environment, evaluation methods, and the most advanced methods that need to be analyzed and compared are given. Then, we verify the effectiveness of our approach through the verification of sub-edge servers and the federated training of edge collaboration.

### 5.1. Environment Settings

We use the PyTorch deep learning framework to implement all experiments for edge computing intelligent services on a 3090 GPU. We consider an innovative service system with 10-edge devices. These edge devices train a local model and then send it to an aggregator *S*, which combines them using FedAvg.

#### 5.1.1. Datasets and Models

We use Cifar10 as our image classification task, training a global model with 20 participants, with ten randomly selected per round. We use the CIFAR10 dataset for evaluation. The CIFAR10 dataset is a dataset of 60,000 images. Each photo is a 32 × 32 color photo, and each pixel includes three RGB values, ranging from 0 to 255. CIFAR10 contains ten different categories, namely ’airplane’, ’automobile’, ’bird’, ’cat’, ’deer’, ’dog’, ’frog’, ’horse’, ’ship’, and ’truck.’ 50,000 images were divided into trainsets, and the remaining 10,000 images belonged to test sets. We adopt RestNet18 CNN model training. Table 1. Detailed information about the datasets and models.

#### 5.1.2. Attack Details

Our experimental environment includes one cloud server, ten edge nodes, and intelligent services connected to multiple clients below. We randomly and uniformly distribute the dataset to 10 edge devices. The edge device employs a cross-entropy loss function and stochastic gradient descent to train the local model. Randomly select clients to plant backdoor triggers and train local models. We use the standard neural network training method model to train a Wide ResNet18 model on the corresponding poisoning data set by solving Equations (9) and (12). Each model is trained according to the requirements we set. We use random gradient descent (SGD) to train 80 reverse models. The initial learning rate of CIFAR-10 is 0.1, the weight attenuation is $10^{-4}$, and the momentum is 0.9. The backdoor attack target tag is set to 0.

#### 5.1.3. Evaluation Metrics

To illustrate our proposed label poisoning attack method, we evaluate it with the following 3 objectives. We mainly evaluate the performance of our defense framework by backdoor attack success rate (ASR) and main task accuracy (Classification Accuracy (CA) on a clean testset), where ASR is the proportion of target samples that the attacker misclassifies as the specified label. An effective defense against backdoors needs to maintain model performance on benign samples while reducing ASR.

#### 5.1.4. Attack Scenario

We tried a label inversion attack [reverse labels to cats before training] [39], as well as a clean label attack (CL) [13], constraint and scaling [14], and a distributed DBA backdoor attack [27]. We set up these attack algorithms according to the open-source code of the original text.

#### 5.1.5. Defense Methods

We compare our layered defense method with state-of-the-art defense methods [20,40]. FL-Defender [41] is a defense framework that estimates the sufficient amount of noise to be injected to ensure the elimination of backdoors. FoolsGold is a robust aggregation algorithm that detects model updates submitted by participants.

### 5.2. Experimental Results

In this section, we mainly verify the defense effect proposed by us from centralized backdoor attacks and distributed backdoor attacks and make a comparative analysis with the most advanced defense methods.

#### 5.2.1. Effectiveness of Layered Defense

We evaluate the effectiveness of our layered defense by implementing different classical backdoor attack methods. These attack methods include constrain-and-scale [14] and clean label attacks [13].

The results are shown in Table 2. We can see from the table that our layered defense framework can defend well against different types of backdoor attacks. Compared with the most advanced FL-Defender [21], the defense effect is almost identical or even better. Our main task model accuracy (CA) is comparable to the model accuracy of clean dataset training, reaching more than 90%. Our attack success rate is reduced to 0.08%. In addition, more types of backdoor attacks can be defended, including anti-label inversion attacks, feature collision attacks, dynamic backdoor attacks, and clean-label attacks. However, FL-Defender [21], FoolsGold [28] are not a good defense against dynamic and clean-label backdoor attacks. For example, the accuracy of the main task drops to 63.5%, and the attack success rate of constrain-and-scale [14] is 100%.

#### 5.2.2. The Necessity of Edge Model Self-Distillation

In the intelligent service of edge computing, we want edge nodes to train as clean a model as possible on toxic data. In this section, we analyzed the relationship between the strong correlation between the backdoor sample breaking at the edge and the backdoor target class and distillation. Table 3 shows our performance results with the cifar10 data after model breaking correlation and distillation. We can see that, after removing the correlation between backdoors at edge nodes, the accuracy of the model is seriously impaired, and the CA of the local model is only 50.32%. After distillation, the CA of the edge node can be improved to the precision before the model. Therefore, it is necessary to perform distillation at the edge nodes.

#### 5.2.3. The Necessity of Cloud Server Defense

Since we do not restrict the attacker’s actions, the attacker can perform backdoor attacks by manipulating model updates, so we have to defend cloud servers again. While more tedious, we can ensure a secure model in the event of an attack. On the server side, in order to prevent the adversary from achieving its attack goals, the impact of backdoor model updates must be removed so that the aggregated global model does not reveal backdoor behavior. In this part, we mainly verify the effect of backdoor defense only on the server side. The results are shown in Table 4.

We can see from Table 4 that the defense effect of only the server side is also good. In the backdoor defense of constrain-and-scale [14], the ASR is only 3.49%, while the performance of CA is almost the same as that of the original model, reaching 81.68%. Although the ASR after distillation increased by 1%, the accuracy of the main task also increased by 2%. Therefore, our backdoor defense framework can not only facilitate the security management of edge computing intelligent service scenarios, but also resist backdoor attacks at any stage.

## 6. Conclusions and Future Work

To facilitate the security management of edge intelligent services, we propose a layered backdoor defense mechanism for edge computing intelligent service scenarios. On the edge side, the backdoor data are detected and eliminated first. Self-attention distillation is performed before the model update is sent to the cloud server to improve the accuracy and security of the model. In the cloud server, we conduct two backdoor update detections and propose and execute an attentional self-distillation mechanism to recover the model’s performance Through test evaluation, we can see that our method is comparable to or even better than the state-of-the-art backdoor attack defense techniques. Our defense framework reduces the ASR to below 10%, and the self-attention distillation operation improves the model’s accuracy by nearly 2%, approaching the performance of the clean model. In addition, our defense framework is closer to the actual needs of edge intelligent computing security management.

Although our method is resistant to backdoor attacks from both sides, we achieve good results. However, our approach is a staged defense, which takes more time while meeting the security requirements of edge service computing. It needs to meet the defense procedures established by most researchers, and we may not be able to defend against multiple backdoor attacks effectively. The defense phase takes longer because we add self-extracting model training when the customer’s first model training area stabilizes. On the server side, we judge not only the model update but also conduct self-distillation training, which makes training the entire edge computing service take a long time. In the future, we will further optimize the edge intelligent computing field to quickly and safely resist backdoor attacks. On the one hand, we defend against several different backdoors. On the other hand, we upgrade and optimize the defense process and time.

## Figures and Tables

**Figure 1 sensors-23-01052-f001:**
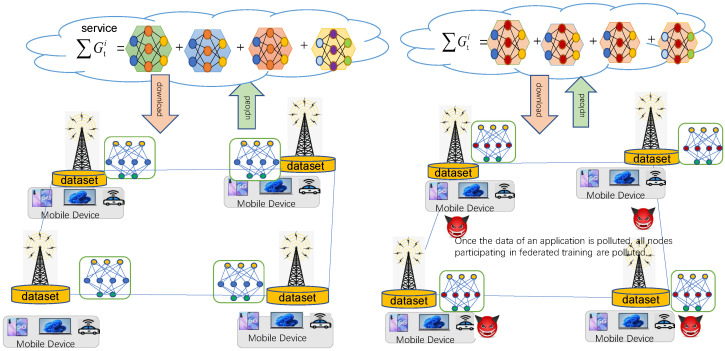
Backdoor attack of intelligent services in edge cooperative computing. Suppose that one of the edge nodes is an attacker who trains the edge model by training samples embedded with backdoor triggers locally and sends its model updates to the cloud server. The cloud server will distribute the updated models of all participants after aggregating them and will spread the embedded attack mode to all edge nodes.

**Figure 2 sensors-23-01052-f002:**
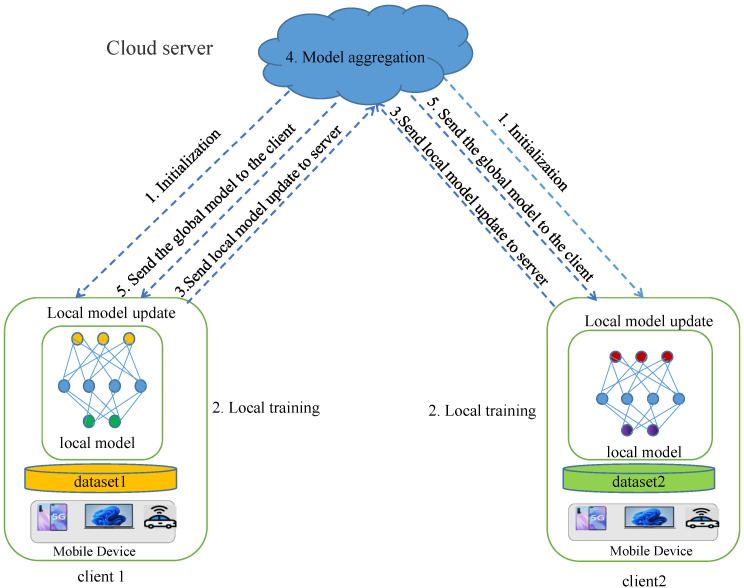
Training process of federated learning.

**Figure 3 sensors-23-01052-f003:**
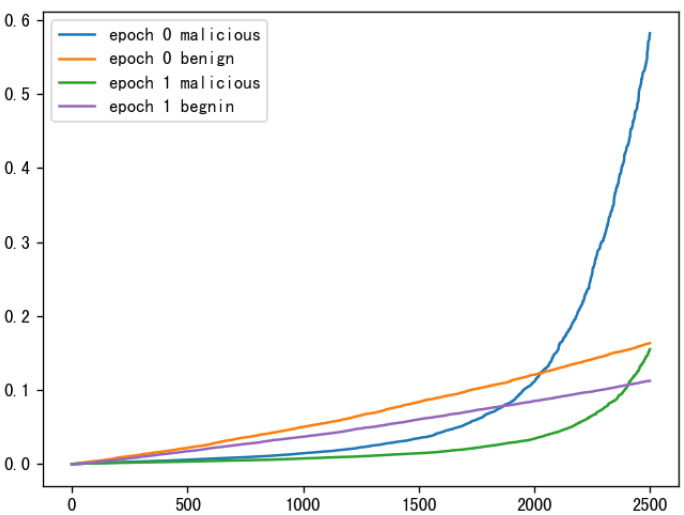
Training process of federated learning.

**Table 1 sensors-23-01052-t001:** Comparative analysis of attacks with different attack methods.

Dataset	Labels Number	Training Samples	Testing Samples	Model Structure
CIFAR-10	10	50,000	10,000	ResNet-18

**Table 2 sensors-23-01052-t002:** The attack success rate (ASR%) and clean accuracy rate (CA%) of three backdoor defense methods against seven backdoor attacks, including Label flip attacks, clean label backdoor attacks, Constrain-and-scale. None means the training data are completely clean.

Dataset	Types	No Defense		FL-Defender [21]		FLPA-SM [41]		Layered Defense (Ours)	
		ASR	CA	ASR	CA	ASR	CA	ASR	CA
Cifar-10	No Attack	0%	82.16%	0%	80.0%	0%	80.01%	0%	82.16%
	Label flip attack [39]	100%	74.35%	6.72%	76.1%	6.32.0%	52.31%	4.98%	80.01%
	clean label attack [13]	81.67%	88.43%	68.22%	64.71%	82.06%	63.55%	8.22%	75.98%
	Constrain-and-scale [14]	99.56%	81.85%	63.45%	72.34%	81.56.0%	52.37%	3.44%	80.26%

**Table 3 sensors-23-01052-t003:** The attack success rate (ASR%) and clean accuracy rate (CA%) of three backdoor defense methods against seven backdoor attacks, including label flip attacks, clean label backdoor attacks, and constrain-and-scale. None means the training data are completely clean.

Dataset	Types	No Defense		Edge Defense		Edge Distillation	
		ASR	CA	ASR	CA	ASR	CA
Cifar-10	No Attack	0%	82.16%	0%	78.31%	0%	82.01%
	Label flip attack [39]	99.3%	74.35%	6.31%	58.90%	6.33%	80.22%
	clean label attack [13]	81.67%	88.43%	8.77%	55.33%	10.96%	79.89%
	Constrain-and-scale [14]	99.56%	81.85%	2.71%	50.32%	5.44%	81.33%

**Table 4 sensors-23-01052-t004:** Global Model Only Defenses.

Datdset	Type	Server Defence		Server Defense Distillation	
		ASR	CA	ASR	CA
Cifar10	No attack	0%	82.16%	-	-
	Constrain-and-scale [14]	2.64%	79.37%	3.49%	81.68%

## Data Availability

The data set we use is the Cifar10 public data set, which can be accessed through https://www.cs.toronto.edu/~kriz/cifar.html download.

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
