# Peer review of "Edge-Cloud Collaborative Defense against Backdoor Attacks in Federated Learning"

_sensors, 2023, doi:10.3390/s23031052_

Round 1

Reviewer 1 Report

The work entitled “Edge-Cloud Collaborative Defense against Backdoor Attacks in Federated Learning” proposes a layered defense framework for edge computing intelligent services with the ABL and attention self-distillation mechanism, and adopts the attention distillation mechanism to get a clean global model to ensure the accuracy of the initial value of the next round of model training. The experimental result is practical and feasible.

The quality of the submission is good and the paper is well organized. However, I have following comments:

1. Part of citations in the manuscript are not correctly given, please revised them.

2. The authors are suggested to list their notations and symbols in one table, and give their corresponding explanations.

3. Figure captions are too short, please include more explanations. Moreover, Figure 1 is indistinct.

4. The authors are suggested to illustrate the proposed approach’s limitations and future works in detail.

5. The titles of subsection 2.1 and 2.5 are same. Please check it.

Reviewer 2 Report

Backdoor attack is a big challenge for federated learning. This manuscript proposed a layered defense method for backdoor attack. This approach integrates the detection ability from two perspectives on both edge side and cloud side.   

In my opinion this manuscript is well written, but I still have some suggestions as follow:

1). More details on the defense performance comparison with existing other methods. 

2). The language should be improved. The grammar of the article is not very smooth in some places and needs to be strengthened.

3). What is the relationship of edge defense and pre-aggregation defense.

Author Response

Please see the Appendix.

Reviewer 3 Report

Federated learning is widely used in the IoT scene of edge-computing intelligent services. Due to the limited bandwidth and unstable network connections of edge computing intelligence services, it is crucial to detect whether the data of edge nodes is contaminated in time. This paper proposes a layered defense framework (including edge defense and cloud defense) for edge computing intelligent services. 

The presentation style of the manuscript must be revised definitely. It need a through revision. Therefore, the paper should be revised by considering the following issues:

MAJOR ISSUES

+ The number of references are insufficient. The related work and bibliography should be improved.

+ Most of the references in this paper are mostly recent publications (within the last 5 years) and relevant. On the other hand, the bibliography should be improved by adding most recent references.

+ Introduction section should be rewritten to improve it.

+ The main contributions of the paper should be clearly given as a separate subsection in the introduction section.

+ The organization of the paper should be clearly given as a separate subsection in the introduction section.

+ In page 5, Section 1.3 includes citation errors like "Nguyen[? ] et al." and "Mi[? ] et al." Similarly, page 7, Section 3.2 includes "the ABL method[? ]" 

+ “Related Work” should be given as a separate section (as Section 2 before Section 3. Background).

+ “Problem Definition and System Model” should be given more clearly.

+ Equation (2) should be rewritten clearly. What is the term " c´  "?

+ The equation at the end of Subsection 2.3 should be given as an equation instead of giving in the text. What is "ffi"?

+ The proposed scheme performs well. The motivation behind it should be explained better.

+ The figures/schemes are generally clear. They show the data properly. It is not difficult to interpret and understand them. On the other hand, Figure 1 should be explained better by adding more information to its caption.

+ Section 4. Experimental Evaluation should be definitely improved. Many more figures should be given in the numerical results section. Figures should be clearly explained, especially in the text/main body of the paper.

+ Preamble information is required between section"1. Background" and subsubsection "1.1. Federated Learning".

+ Preamble information is required between subsection"4.2. Experimental Results" and subsubsection "4.2.1. Effectiveness of Layered defense"

+ The conclusion should be improved by giving the key results and main contributions more clearly.

+ Future work part should be given in the conclusion section.

MINOR ISSUES

+The grammatical errors and typos should be fixed.

+The authors should adjust the (section) counter as "0" instead of the default value "-1".

"Table 1:" should be removed from the caption of Table 1.

+The references in the bibliography should be given in the same style. The following link should be checked: https://www.mdpi.com/authors/references 

Round 2

Reviewer 2 Report

My comments are responded to in this revised manuscript.  I have no more comments.

Author Response

Thank you very much for your comments.

Reviewer 3 Report

The paper is generally well written. The authors addressed my comments on the previous version of the paper considerably. On the other hand, they should consider the following issues:

+ Table 3 should not exceed page margins.

+ The preamble information between Section "4. Problem Setting and Objectives" and subsection "4.1. Problem definition" should be improved.

+ The preamble information between Section "5. Proposed Method" and subsection "5.1. Overview " should be improved.

+ The preamble information between Section "6. Experimental Evaluation" and subsection "6.1. Environment settings".

+ The  paper should also consider deep-learning-assisted RF fingerprinting approach (possibly in its introduction and related work sections) as a solution for malicious attacks including Sybil attacks, spoofing attacks, etc. Some of the recent papers on RF Fingerprinting can be listed as follows.

- G. Reus-Muns, D. Jaisinghani, K. Sankhe and K. R. Chowdhury, "Trust in 5G Open RANs through Machine Learning: RF Fingerprinting on the POWDER PAWR Platform," GLOBECOM 2020 - 2020 IEEE Global Communications Conference, 2020, pp. 1-6, doi: 10.1109/GLOBECOM42002.2020.9348261.

- Ceren Comert, Michel Kulhandjian, Omer Melih Gul, Azzedine Touazi, Cliff Ellement, Burak Kantarci, and Claude D'Amours. 2022. Analysis of Augmentation Methods for RF Fingerprinting under Impaired Channels. In Proceedings of the 2022 ACM Workshop on Wireless Security and Machine Learning (WiseML '22). Association for Computing Machinery, New York, NY, USA, 3–8. https://doi.org/10.1145/3522783.3529518

- O. M. Gul, M. Kulhandjian, B. Kantarci, A. Touazi, C. Ellement and C. D'Amours, "Fine-grained Augmentation for RF Fingerprinting under Impaired Channels," 2022 IEEE 27th International Workshop on Computer Aided Modeling and Design of Communication Links and Networks (CAMAD), 2022, pp. 115-120, doi: 10.1109/CAMAD55695.2022.9966888.

+ The authors should explain the motivation of the paper and proposed approach more by considering the abovementioned papers.

+ Minor spell-check is required.

+ The figures should be enlarged.

Author Response

REVISION OF MANUSCRIPT

TITLE: Edge-Cloud Collaborative Defense against Backdoor Attacks in Federated Learning” AUTHORS: Jie Yang , Jun Zheng,  Haochen Wang,Jiaxing Li1,, Haipeng Sun, Weifeng Han, Nan Jiang , and Yu-an Tan

Dear Respective Editor-in-Chief, Guest Editors, and Reviewers,

We would like to thank the respective Reviewers, Guest Editors, and Editor-in-Chief for their valuable comments on the manuscript along with their suggestions of improvements. These comments and suggestions have been considered when preparing the revised version of the manuscript. The remainder of this response letter explains how we have handled the reviewers’ comments and implemented their suggestions.

We hope that the changes and improvements we made in the manuscript will satisfy the requirements of all the Reviewers, Guest Editors, and Editor-in-Chief, which will lead to accepting the manuscript for publication in Journal of Sensors..

Best regards,

All the Authors

********************** Reviewer #3*****************

Reviewer#3, Concern # 1: Table 3 should not exceed page margins.

Author response:

Thank you very much for your suggestion. According to your suggestion, we modified the margins of Table 3.

And the modification has been highlighted in the manuscript.

Reviewer#3, Concern # 2: The preamble information between Section "4. Problem Setting and Objectives" and subsection "4.1. Problem definition" should be improved. 

Author response:

Thank you very much for your suggestion. According to your suggestion, we have rewritten the information between Section "4. Problem Setting and Objectives" and subsection "4.1. Problem definition".

The modification can be seen in the revised manuscript.  

Reviewer#3, Concern # 3: The preamble information between Section "5. Proposed Method" and subsection "5.1. Overview " should be improved.

Author response:

Thanks a lot for your question. According to the problem, we have rewritten the information between Section "5. Proposed Method" and subsection "5.1. Overview ".

we have rewritten the introduction.

Reviewer#3, Concern # 4: The paper should also consider deep-learning-assisted RF fingerprinting approach (possibly in its introduction and related work sections) as a solution for malicious attacks including Sybil attacks, spoofing attacks, etc. Some of the recent papers on RF Fingerprinting can be listed as follows.- G. Reus-Muns, D. Jaisinghani, K. Sankhe and K. R. Chowdhury, "Trust in 5G Open RANs through Machine Learning: RF Fingerprinting on the POWDER PAWR Platform," GLOBECOM 2020 - 2020 IEEE Global Communications Conference, 2020, pp. 1-6, doi: 10.1109/GLOBECOM42002.2020.9348261. - Ceren Comert, Michel Kulhandjian, Omer Melih Gul, Azzedine Touazi, Cliff Ellement, Burak Kantarci, and Claude D'Amours. 2022. Analysis of Augmentation Methods for RF Fingerprinting under Impaired Channels. In Proceedings of the 2022 ACM Workshop on Wireless Security and Machine Learning (WiseML '22). Association for Computing Machinery, New York, NY, USA, 3–8. https://doi.org/10.1145/3522783.3529518 - O. M. Gul, M. Kulhandjian, B. Kantarci, A. Touazi, C. Ellement and C. D'Amours, "Fine-grained Augmentation for RF Fingerprinting under Impaired Channels," 2022 IEEE 27th International Workshop on Computer Aided Modeling and Design of Communication Links and Networks (CAMAD), 2022, pp. 115-120, doi: 10.1109/CAMAD55695.2022.9966888. 

Author response:

Thank you very much for your suggestion. According to your suggestion, we have added the introduction and reference of Sybil attacks and other works in the relevant work section. In the introduction, we have added references to the above three articles.

The modification can be seen in the revised manuscript. 

Reviewer#3, Concern # 5: + The authors should explain the motivation of the paper and proposed approach more by considering the abovementioned papers.

Author response:

Thank you very much for your suggestion. we have added references to these three articles in the introduction, although we do not think these three articles are relevant to the subject of this article..

The modification can be seen in the revised manuscript.

Reviewer#3, Concern # 6: Minor spell-check is required. Author response:

Thank you very much for your suggestion. According to your suggestion, according to your opinion, we carefully checked the grammar of the full text

.  The modification can be seen in the revised manuscript.

Reviewer#3, Concern # 7: The figures should be enlarged.

Author response:

Thank you very much for your suggestion. According to your suggestion, we have enlarged both Figure 1 and Figure 2.

 The modification can be seen in the revised manuscript.
